# THE CURIOUS CASE OF REPRESENTATIONAL ALIGNMENT: UNRAVELLING VISIO-LINGUISTIC TASKS IN EMERGENT COMMUNICATION

**Tom Kouwenhoven**[1], **Max Peeperkorn**[2], **Bram van Dijk**[1], **Stephan Raaijmakers**[3], **Tessa Verhoef**[1]

[1]Leiden Institute of Advanced Computer Science, Leiden University, Netherlands
[2]School of Computing, University of Kent, United Kingdom
[3]Leiden University Centre for Linguistics, Leiden University, Netherlands
`t.kouwenhoven@liacs.leidenuniv.nl`

## ABSTRACT

Natural language has the universal properties of being compositional and grounded in the real world. A popular method to investigate the emergence of linguistic properties is by simulating emergent communication setups with deep neural agents in referential games. Despite growing interest, experiments have yielded mixed results compared to similar experiments addressing linguistic properties of human language. Here we address *representational alignment* as a potential contributing factor to these results. Specifically, we investigate the alignment between agent image representations and between agent representations and the input images. We first revisit and confirm that the emergent language in the common referential game does not appear to encode conceptual visual features, since agent image representations drift away from the input whilst inter-agent alignment increases. We further find a strong relationship between inter-agent alignment and topographic similarity, a common metric for compositionality, and address its consequences. We then introduce an alignment penalty that results in equivalent communicative success but prevents representational drift. Overall, we show critical differences between emergent solutions from humans and neural agents and highlight the importance of representational alignment in simulations of language emergence.

## 1 INTRODUCTION

Human language has unique properties that make it a powerful tool for communication. A well-known property is compositionality: the ability to combine meaningful words into more complex meanings (Hockett, 1959). The emergence of compositionality is studied extensively in the field of language evolution through human experiments (e.g., Selten & Warglien, 2007; Kirby et al., 2008; 2015; Raviv et al., 2019a). An important finding from this field is that the unique nature of human language can be explained as a consequence of biases for simplicity and expressivity imposed during continuous language learning and use (Smith, 2022). Besides experimental studies, computational simulations have also been used to study the emergence of linguistic properties (e.g., de Boer, 2006; Steels & Loetzsch, 2012), and have seen a rising interest in the field of computational linguistics (Lazaridou & Baroni, 2020). Here, the degree of compositionality in the emergent protocols is commonly measured through topographic similarity (TOPSIM; Brighton & Kirby, 2006). It measures the topographic relation between meanings and signals, conceptually it gauges whether similar meanings map to similar signals. This metric was introduced in simulations of emergent communication by Lazaridou et al. (2018) and has been used in a large body of work since. Yet, it is still unclear how the emergence of linguistic properties in simulations should be interpreted seeing that language protocols used among artificial agents often show critical mismatches with known properties of human languages (Galke et al., 2022; Lian et al., 2023). It is therefore crucial to obtain deeper insight into referential games in the language learning setting (Rita et al., 2022).

A possible explanation for these mismatches could stem from representational alignment, the degree of agreement between the internal representations of two information processing systems (Sucholutsky et al., 2023). To the best of our knowledge, representational alignment was first, and only, reported

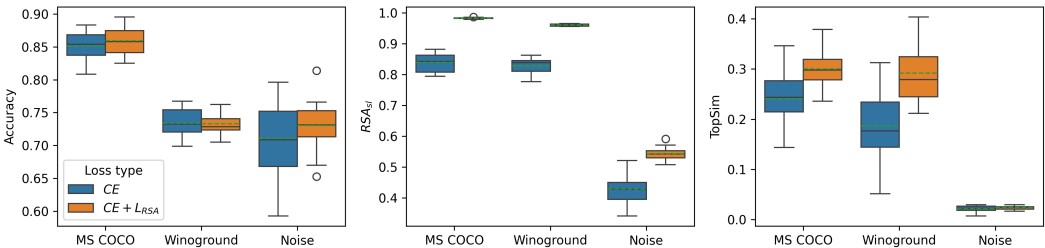

Figure 1: Left: communicative performance on three datasets across 15 seeds, using the best-performing parameters from our parameter sweep. Middle: inter-agent representational alignment (RSA) between the agents. Right: topographic similarity (TOPSIM) between the image input and the produced messages. Dashed green lines indicate averages.

by Bouchacourt & Baroni (2018), who measured the degree to which communicating agents aligned their internal image interpretations (inter-agent alignment) by performing Representational Similarity Analysis (RSA; Kriegeskorte et al., 2008). Using RSA (§3), they showed that agents establish successful communication artificially by aligning their internal image representations while *losing* any relation to the images presented (image-agent alignment), enabling communication about noise input even though they were trained on real images. As such, their language protocol captured not conceptual properties of the objects depicted in pictures but most likely focused on spurious image features. While inter-agent alignment is not a problem per se, losing image-agent alignment is problematic for two reasons. Firstly, for simulations of emergent communication to be informative of human language emergence, agent image representations must be grounded in what is represented in the images. Only then can we deduce *what* the agents communicate about and investigate linguistic properties or their ability to generalise to novel concepts. Secondly, emergent communication setups have been proposed to fine-tune pre-trained (vision-)language models to enhance machine understanding of natural human language (Lazaridou & Baroni, 2020; Lowe et al., 2020; Steinert-Threlkeld et al., 2022; Zheng et al., 2024). Here, representations must maintain substantial alignment with the image to maintain mutual understanding with humans.

Representational alignment, however, did not receive the necessary attention since a host of papers appeared *after* the findings by Bouchacourt & Baroni in which results on referential games were reported without taking RSA into account (e.g., Lazaridou et al., 2018; Guo et al., 2019; Li & Bowling, 2019; Ren et al., 2020; Chaabouni et al., 2020b; Dagan et al., 2021; Mu & Goodman, 2021; Chaabouni et al., 2022). Admittedly, some use attribute-value objects and not real images as input. But importantly, the problem of inter-agent alignment is *agnostic* to the input type and can *always* occur when agents map inputs into an agent-specific representation, which is the case for almost all simulations. Although this warrants further analysis of earlier results, the field is already employing referential games in even more complex simulations with real images (e.g., Dessi et al., 2021; Chaabouni et al., 2022; Mahaut et al., 2024).

This work addresses the understudied alignment problem in standard referential game setups used in emergent communication. We train Reinforcement Learning (RL) agents equipped with a recent vision module (DinoV2; Oquab et al., 2024) to communicate about images. In addition to evaluating the agents on MS COCO (Lin et al., 2014) image pairs, we evaluate on noise pairs and image pairs sourced from the Winoground dataset (Thrush et al., 2022). The latter is explicitly created to gauge visio-linguistic compositional reasoning abilities of vision and language models. We first confirm that effective communication in the referential game relies on inter-agent alignment and then move on to our contributions. First, we find a strong correlation between the degree of inter-agent alignment and the TOPSIM metric. Our second contribution consists of a solution to the alignment problem (§3) by including an alignment penalty term to the loss, resulting in equivalent communicative success and higher TOPSIM whilst ensuring that the agents communicate about images instead of spurious features (Figure 1). We then argue to start evaluating emergent communication protocols on more strict tasks that directly target the intuition behind popular metrics to obtain a clearer understanding of the protocols. Overall, our results highlight the importance of representational alignment in simulations of language emergence, and underscore the need to better understand the contradictions between human and artificial language emergence.

## 2  BACKGROUND

Most research in simulating emergent communication is modelled after the Lewis signalling game (Lewis, 1969) with a speaker and a listener agent. The speaker observes an environment state (e.g., an image) and sends a signal to the listener who acts based on this signal. In the case of the referential game, this means selecting a target among multiple distractors. Both agents are rewarded for successful communication, meaning the listener points to the target object. The solution of this game requires the agents to have a shared protocol (i.e., an artificial language) which typically emerges when the agents learn based on trial and error over multiple games. This is similar to how language learning and use for humans impose constraints like pressures for learnability and compression that shape our language design (Kirby et al., 2014; 2015). Importantly, the emergent language in this setup is also shaped by biases resulting from, for example, the agent architecture, loss function, and learning protocol Rita et al. (2022). In the current work, we use the referential game: a variant of the Lewis signalling game which is extensively used in linguistic and cognitive studies to explore language evolution (e.g., Steels & Loetzsch, 2012; Kirby et al., 2015; Lazaridou et al., 2017; Kottur et al., 2017; Lazaridou et al., 2018; Kharitonov et al., 2020; Chaabouni et al., 2022).

An important challenge in emergent communication is that artificial learners often do not behave the same way as human learners in experimental settings. Some emergent protocols do not follow Zipf's law and thus are anti-efficient unless pressures for brevity are introduced (Chaabouni et al., 2019a), others do not show the word-order vs. case-marking trade-off found in human languages (Chaabouni et al., 2019b; Lian et al., 2021). It has been suggested to introduce communicative (e.g., alternating speaker/listener roles) and cognitive (e.g., memory) constraints (Galke et al., 2022) and use more natural settings to promote more human-like patterns of language emergence with neural agents (Kouwenhoven et al., 2022). An example of such work, investigating the word-order vs. case-marking trade-off, has succeeded in replicating this trade-off for neural learners (Lian et al., 2023). Their setup differs from other work in that agents first learn a miniature language via supervised learning, and then optimise it for communication via RL, resulting in emergent languages that share linguistic universals with human language. Yet, their work is based on the reconstruction game, not the referential game, the topic of this paper.

To enhance understanding of emergent communication in the Lewis game, Rita et al. (2022) decomposed the standard objective in Lewis games into two key components: a co-adaptation loss and an information loss. In doing so, they shed light on potential sources of overfitting and how they might hinder the emergence of structured communication protocols. They demonstrated that desired linguistic properties (e.g., compositionality and generalizability) emerge when they control the listener's ability to converge to the speaker agent (i.e., control for overfitting on the co-adaptation loss). While the co-adaptation loss has parallels to inter-agent alignment, their work does not address the alignment between the agents' image representation and the input features, which we deem crucial in developing grounded communication protocols.

Another challenge in emergent communication is the disentanglement of the underlying meanings of emergent languages. Earlier research suggested that the meanings agents assign to symbols capture general conceptual properties of the objects in images rather than low-level visual properties (Lazaridou et al., 2017). However, as previously mentioned, follow-up work from Bouchacourt & Baroni (2018) showed this is not always the case, as agents align their agent-specific image representations and do not share a language that captures conceptual properties depicted in the images. Moreover, agents lost any sense of meaningful within-category variation where two similar objects in human perception (e.g., two avocados) were observed as maximally dissimilar for the agents. Although this is a pressing matter that needs to be addressed *before* continuing with multi-modal setups, to the best of our knowledge, there has been little attention to their results apart from testing whether trained agents can communicate about noise (Dessi et al., 2021; Mahaut et al., 2024).

## 3  REPRESENTATIONAL ALIGNMENT

Representational alignment is the degree of agreement between the internal representations of two information processing systems, whether biological or artificial (Sucholutsky et al., 2023). Even though representational alignment is widely recognised in cognitive science, neuroscience, and

machine learning (Sucholutsky et al., 2023), it has not seen much interest in the field of emergent communication, except for the work by Bouchacourt & Baroni who analysed the referential game using RSA. This metric measures the alignment between two sets of numerical vectors, for example, image embeddings and agents' representations thereof. In practice, it is calculated by taking the pairwise (cosine) distances between vectors of a set and calculating the Spearman rank correlation between these distances.

Representational alignment is also operationalised using RSA in this paper. Given the speaker image representations $r_s$ of the DinoV2 input embeddings $i$ and $r_l$ as the same images represented in the listener representation space, we compute the pairwise cosine similarity between the representations for the speaker $s_s$ and for the listener $s_l$ and calculate Spearman's $\rho$ between $s_s$ and $s_l$. As such, this measures the degree of inter-agent alignment ($\text{RSA}_{sl}$) between image representations $s_s$ and $s_l$, relative to their input. Additionally, we use it to measure image-agent alignment between the speaker/listener image representations and the DinoV2 embeddings ($\text{RSA}_{si}$ and $\text{RSA}_{li}$). Importantly, alignment is *agnostic to the type of input*, being either images or attribute-value objects and can always happen when inputs are projected onto agent-specific representations.

Intuitively, a high inter-agent $\text{RSA}_{sl}$ value can be interpreted as agents with *similar* representations for similar images. Importantly, this can have two causes: both agents' image representations either *maintain* a relation to the image input, or *lose* this relation. While the former is desirable, the latter means that the agents are not communicating about the same high-level image features but are likely communicating about spurious features. A low $\text{RSA}_{sl}$ value entails that the agents have developed *different* interpretations for the same image. While this may well be similar to the question of whether people have different perceptual experiences of colour (Locke, 1847), in the case of emergent communication, the agents should develop a grounded vocabulary with overlapping concept-level properties if we wish machines to have more natural understanding of human language. Whereas Bouchacourt & Baroni (2018) used RSA to indicate the alignment problem, we use it as 1) a metric to re-assess their findings in whether the agents create messages based on the image features and 2) as an auxiliary loss to mitigate the alignment problem and ensure that the agents communicate about image features.

## 4 METHODS

The standard referential game is used as provided by the EGG framework (Kharitonov et al., 2021). Doing so ensures that our findings are representative of this setup and not specific to design choices. The game is implemented as a multi-agent cooperative RL problem where a speaker and a listener communicate to discriminate a target image from two shuffled distractor images. The speaker receives a target $t$ and generates a message $m$ of at most length $L$, using vocabulary $V$. Importantly, the messages and symbols have no a priori meaning but are assumed to obtain meaning and become grounded during the game. The meaningful symbols are ideally combined in a structured manner to create compositional messages that express more complex meanings. Using message $m$, the listener guesses the target $\hat{t}$. Communicative success is defined as $\hat{t} = t$, meaning that the listener has correctly identified the target image among the candidate images.

### 4.1 AGENTS

The agents contain a language and a vision module. The latter consists of a frozen pre-trained visual network (DinoV2) and a trained agent-specific representation part. While it is difficult to know exactly what conceptual image features are present in DinoV2 embeddings, they provide rich enough features for semantic segmentation (Oquab et al., 2024), which is similar to the agents' task. The language module is trained from scratch.

*The speaker* performs a linear transformation on the image embeddings to obtain its agent-specific image representation $r_s$ followed by batch normalisation. Its language module embeds this representation and passes it through a single-layer Gated Recurrent Unit (GRU; Cho et al., 2014) that spells out messages to describe the target. *The listener* receives the message and the distractor images. It encodes the message into an embedding using another single-cell GRU layer. Additionally, a listener image representation $r_l$ is obtained for each image by performing a linear transformation followed by batch normalisation. Subsequently, temperature-weighted (temperature defaults to 0.1) cosine scores

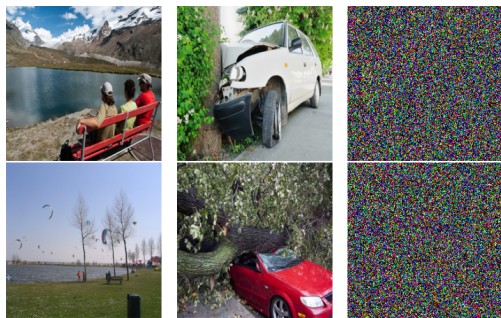

Figure 2: Exemplar pairs used for evaluation. Left: an image pair from MS COCO. Middle: A Winoground example. Right: A Gaussian noise pair. All images are cropped for display purposes.

construct a multi-modal representation between the image and message representation (Dessi et al., 2021), where a higher probability should be assigned to the target image.

## 4.2 OPTIMISATION

Communicative success ($\hat{t} = t$) is used to optimise the trainable parameters of both agents. The listener is optimised to minimise cross-entropy ($ce$) loss using stochastic gradient descent, amounting to supervised learning. The $ce$ loss is calculated over the listeners' target distribution, thus providing direct pressure for communicative success. At inference, the candidate image with the highest probability is chosen as the target $\hat{t}$. The gradients required to optimise the speaker are calculated using the REINFORCE (Williams, 1992) update rule as each generated symbol must be assigned a loss. Following common practice (Rita et al., 2024), entropy regularisation (Mnih et al., 2016) is added to the loss to maintain exploration in message generation.

In addition to the conventional $ce$ loss, we introduce an alignment loss ($ce + \text{RSA}$) that includes an alignment penalty term to enforce high inter-agent and image-agent alignment. The term

$$L_{\text{RSA}} = (1 - \text{RSA}_{sl}) + (1 - \text{RSA}_{si}) + (1 - \text{RSA}_{li})$$

is added to the $ce$ loss with equal importance. We use torchsort (Blondel et al., 2020) to calculate $L_{\text{RSA}}$ such that the entire loss term is differentiable. Importantly, $L_{\text{RSA}}$ is not influenced by communicative success and does not interact with the $ce$ loss (Appendix C). Only adding $\text{RSA}_{sl}$ to the $ce$ loss is not sufficient, as high inter-agent alignment can be achieved while *losing* image-agent alignment (see §3). Including $\text{RSA}_{si}$ and $\text{RSA}_{li}$ intuitively ensures that the agents communicate about the content displayed in the images. In both cases, we train for 30 epochs. The hyperparameters which resulted in the best validation accuracy across 42 different communication channel capacities (Appendix A) were used for our findings (Appendix B).

## 4.3 DATA

The agents are trained to discriminate images from MS COCO but tested on three different datasets (Figure 2) to assess out-of-distribution (o.o.d.) performance.

**MS COCO** – We use a subset of 1200 images from the MS COCO 2017 validation set to train and test the agents using an 80/20 split. To obtain this subset, we first select the categories that contain more than 100 images (12 categories) and subsequently sample 100 images for each supercategory present in the resulting set of images. The distractor images are sampled from the same category to ensure that there is *some* relevance to the target image. Importantly, sampling distractor images is done for each batch, meaning targets have different distractors at each epoch.

**Winoground** – The Winoground dataset (Thrush et al., 2022) was created to assess the visio-linguistic compositional reasoning abilities of vision and language models. Here, we repurpose it as a proxy for the agents' ability to endow in compositional reasoning for image-based settings. The dataset contains 800 images and corresponding captions, comprising 400 Winoground pairs. Image-caption pairs

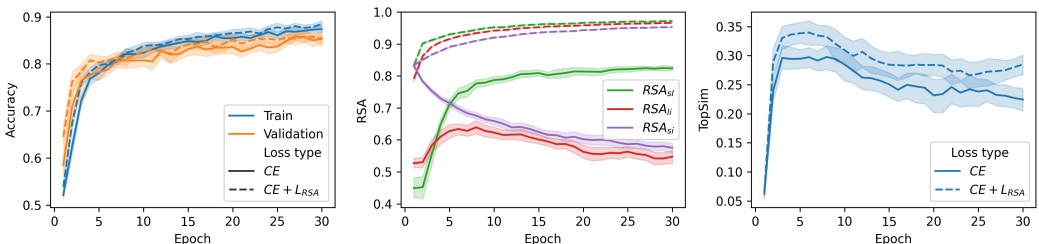

Figure 3: Left: Learning curves for the MS COCO dataset on train and validation data. Middle: Representational alignment between the agents' image representations (green) and between the image features and the sender/listener representations (purple, red). Right: The evolution of topographic similarity (TOPSIM). Data is averaged over 15 seeds, areas indicate the 95% confidence intervals.

were included when the captions share the same words but are of different *compositions*, implying completely different semantics (e.g., "a tree smashed into a car" versus "a car smashed into a tree"). We only use the image pairs, not the captions. Crucially, this task differs from MS COCO since the image pairs are *fixed*, *conceptually similar* and meant to be discriminative if the agents' language allows for compositional reasoning and is grounded in the visual modality.

**Noise** – Following Bouchacourt & Baroni (2018), we test whether agents can communicate about Gaussian noise ($\mu = 0, \sigma = 1$) pairs when trained on real images. This would imply that messages communicate about spurious instead of high-level concept features.

## 4.4 METRICS

The performance of our agents is assessed by communicative success (accuracy) and RSA (§3) measures alignment. The degree of compositionality in the emergent language is assessed through the TOPSIM metric. Other computational metrics for compositionality like positional disentanglement, bag-of-symbols disentanglement (Chaabouni et al., 2020b), or metrics of compositionality that allow for natural language-like variation, e.g., synonymy, homonymy, freedom, or disentanglement (Conklin & Smith, 2023) are not appropriate due to the continuous nature of the image embeddings.

## 5 RESULTS

### 5.1 COMMUNICATIVE SUCCESS

The results are given in Figure 1. Using an emergent language, agents can disambiguate images among MS COCO pairs. Additionally, we observe that agents can communicate about Gaussian noise when trained on real images, confirming previous work (Bouchacourt & Baroni, 2018) and suggesting that the messages convey spurious features rather than concept-level information. Performance on noise is roughly the same on average as on the Winoground pairs, which requires the messages to capture concept-level properties, showing it is difficult to discriminate between strict pairs of conceptually similar images. Compared to MS COCO, the lowered o.o.d. performance is like what is observed in other work (Lazaridou et al., 2018; Conklin & Smith, 2023).

### 5.2 THE ALIGNMENT PROBLEM

The solid lines in Figure 3 (middle) clearly show that inter-agent alignment increases while alignment sensitivity to image features decreases for both agents. In principle, it is not a problem that the agents' image representations align. However, it is problematic when the alignment between the image embeddings and the image representations declines. Ablations across different channel capacities (§A) and pre-trained vision modules (§D) showed that these trends appear consistently and are not influenced by the capacity or type of vision model. This re-confirms that the agents do not learn to extract concept-level information from the image embeddings but instead solve this task differently.

### 5.3 TOPSIM AND ALIGNMENT

Earlier findings show mixed results on the relationship between TOPSIM and generalisation in image-based settings, TOPSIM was either related to generalisation (Chaabouni et al., 2022) or not (Rita et al., 2022). Our results indicate that generalisation and TOPSIM are correlated with both $ce$ ($r = .856$, $p < .001$) and $ce + \text{RSA}$ ($r = .767$, $p < .001$) losses. Meaning that more structured languages enable better communication on unseen validation pairs. Moreover, Figure 4, shows a strong positive relationship between $\text{RSA}_{sl}$ and TOPSIM ($r = .838$, $p < .001$) in the $ce$. This relation is also present in the $ce + \text{RSA}$ setup ($r = .408$, $p = .001$), but is decoupled from TOPSIM given the (very) small spread ($\sigma = .003$) of $\text{RSA}_{ls}$. We do not observe an influence of inter-agent alignment on the number of uniquely produced messages.

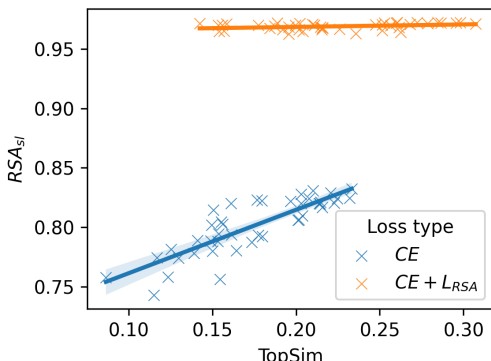

Figure 4: The relationship between TOPSIM and inter-agent alignment ($\text{RSA}_{sl}$) for both loss types.

### 5.4 MITIGATING THE ALIGNMENT PROBLEM

We now focus on the $ce + \text{RSA}$ setup for which we want the agents to maintain alignment with the image embeddings. Figure 3 shows that this is the case: inter-agent alignment *and* agent-image alignment increase during training and remain high at inference. There does not seem to be a benefit for communicative success at inference time (Figure 1). This is because the alignment penalty only forces agents to represent images similarly to the image embeddings and is independent of the cross-entropy loss used to assess the success of communication (Appendix C). In the case of noise images, we still observe above-chance performance, suggesting that communication between the agents still occurs in an artificial manner.

The alignment penalty also leads to increased TOPSIM, indicating a higher level of structure (Figure 3) and strengthens our finding that TOPSIM and inter-agent alignment are related. Suggesting that the observed variations in TOPSIM, whether higher or lower, as noted in previous studies (e.g., Kottur et al., 2017; Chaabouni et al., 2020b), should not be interpreted without considering alignment since they may be attributable to this underlying artefact rather than alterations to the original setup.

When tested on more strict Winoground pairs, communicative success does not improve as a result of using the alignment penalty. Given the correlation between TOPSIM and generalisation (§5.3), this is surprising since the higher degree of TOPSIM should imply that the language is more structured. Moreover, both, $\text{RSA}_{si}$ and $\text{RSA}_{li}$ have not drifted away from the image features. This combination, *in theory*, should be ideal for discriminating image pairs from the Winoground dataset since it was designed to be discriminative with compositional visio-linguistic reasoning. However, in *practice* this is not the case.

## 6  DISCUSSION

In this work, we revisited the representational alignment problem in a common setup used in emergent communication and proposed a solution to this underrepresented problem. We corroborated earlier findings by showing that agents align their image representations and rely on spurious image features

instead of concept-level information (Bouchacourt & Baroni, 2018). We then showed that inter-agent alignment strongly correlates with the commonly used TOPSIM metric. Our solution to the alignment problem involves an alignment penalty that forces the agents to remain aligned with the input features and mitigates the alignment problem without decreasing communicative success. Finally, when agents are tested on more challenging Winoground pairs they maintain reasonable but lower performance while representing images similarly to the image embeddings, instead of relying on spurious features. With this work, we hope that the alignment problem will receive more attention in the field of emergent communication, as is already the case in adjacent fields (Sucholutsky et al., 2023).

It is common practice in simulations of emergent communication to process (visual) inputs into an agent-specific hidden representation and update their weights simultaneously (e.g., Lazaridou et al., 2017; Bouchacourt & Baroni, 2018; Chaabouni et al., 2019a; 2020b; Rita et al., 2022). As such, inter-agent alignment, *irrespective of the input form*, likely happens in other simulations too. This phenomenon is therefore potentially widespread and perhaps the cause for findings that are at odds with experimental findings. While it is not always the case that the representation structure we *expect* to help solve a task will do so (e.g., Montero et al., 2021; Xu et al., 2022), such discrepancies may hinder the use of emergent communication models in developing a more natural understanding of human languages and leave them less suitable for directly simulating language evolution phenomena. Especially if we want machine representations of natural language to align with human representations (Sucholutsky et al., 2023). RSA should therefore be used to rule out, or at the bare minimum report about, representational alignment in the future.

Measuring representational alignment using RSA is similar to how TOPSIM measures the structure in messages. They differ in their inputs but both measure the correlation between pairwise distances, which are metric-agnostic. Crucially, the input makes all the difference, the inputs for RSA are from both agents and are trained independently, whilst TOPSIM only assesses the relation between the fixed inputs and learned output. Despite the similarities, the metrics thus describe different phenomena and are rarely reported simultaneously.

We hypothesise that the relationship between TOPSIM and inter-agent representational alignment is a by-product of the setup, which in essence implies that the listener has to align its representation $r_l$ to the speaker representation $r_s$ (Rita et al., 2022). It has to do so using only the speakers' messages, which are an abstraction of $r_s$. A solution to this problem is to align representations, which eases the listeners' training objective. If the speaker consistently produces structured messages during training, aligning $r_l$ to $r_s$ is easier, thereby causing higher inter-agent alignment. Essentially, this renders TOPSIM to be an *indirect* metric for the rate of alignment, for which $RSA_{sl}$ is a *direct* metric. In the context of learnability, the found relationship between TOPSIM and inter-agent alignment and that alignment always occurs can be seen as reasons for why languages with higher TOPSIM are easier to learn (Li & Bowling, 2019; Cheng et al., 2023). This underscores the need to report inter-agent representational alignment to avoid conclusions drawn about the effect of specific interventions on TOPSIM which may be attributable to inter-agent alignment.

We used the Winoground dataset as a proxy for the agents' ability to endow in compositional reasoning for image-based settings. Good performance on the Winoground dataset requires a grounded vocabulary that can be used to create compositional messages since the objects and their underlying relations need to be described. In general, we suggest to start evaluating simulations of referential games on targeted strict tasks, like probing state-of-the-art vision language models on e.g., visio-compositional (Thrush et al., 2022; Diwan et al., 2022; Hsieh et al., 2023; Ray et al., 2023) or spatial (Kamath et al., 2023) reasoning. Re-purposing such datasets can reveal more directly whether agents develop the attested communicative abilities that are trivial to humans without relying on metrics. Our results illustrate this through another shortcoming of the TOPSIM metric. We observed that agents still struggle with distinguishing pairs of *conceptually similar* Winoground images even though TOPSIM is higher with the alignment penalty. If the language protocol were to communicate concept-level information *and* compositional messages were created, we should not observe this struggle, meaning that the protocols do not enable human-like communicative success.

An important implication of our findings concerns the standard practice of reporting o.o.d. accuracy where the agents are tested on unseen input after training (e.g., Auersperger & Pecina, 2022; Conklin & Smith, 2023). This should inform about the agents' ability to generalise from one dataset (e.g., MS COCO) to another dataset (e.g., the Winoground pairs) much like human language allows us to talk about an infinite number of situations. Crucially, this overlooks the representational alignment

problem in that we do not know *what* the agents are precisely generalising about. This problem can be mitigated with the alignment penalty to assess generalisation more directly. We assess o.o.d. performance on the more challenging Winoground pairs and observe roughly equal accuracy when the alignment penalty is used compared to the $ce$ loss. Interestingly, the o.o.d. performance remains substantially above chance in the $ce + \text{RSA}$ setting. Given that MS COCO is not a dataset for learning to model compositionality, this delineates the limits of what can be achieved qua performance based on MS COCO image features in the Winoground context. Nevertheless, this leaves open the question of above-chance performance on Gaussian noise with the $ce + \text{RSA}$ loss. A tentative explanation is that the higher inter-agent alignment on noise input ($M_{ce} = .428$, $M_{ce+\text{RSA}_{ls}} = .543$, $t = -8.71$, $p < .001$) alleviates part of the problem. To validate this, future experiments should involve controlling the prior distributions of the agents' image encoders by training their vision modules on different data. Doing so ensures that they have to communicate about novel objects and cannot rely on similar representations.

## 7   CONCLUSION

This paper revisits the underrepresented alignment problem present in the well-known referential game often used in simulations of emergent communication. Specifically, we focused on the problem of increasing alignment between agent-image representations in combination with a decreasing alignment between the input and agent representations. We first confirmed that the emergent language in referential games does not appear to encode visual features, since the agents align their image representations while losing connection to the input. We then showed that, in the common setup, inter-agent alignment is related to topographic similarity, and argued that this renders TOPSIM an *indirect* metric of the rate of inter-agent alignment. To further investigate the effects of alignment, we introduced an alignment penalty to mitigate the alignment problem and showed communicative ability on a strict compositionality benchmark. Our findings underscore critical differences between human and artificially emergent solutions within the prevalent referential setup, and highlight the importance of representational alignment and its potential impact on simulations of language emergence. We hope that future work rules out or at least reports about representational alignment.

## 8   LIMITATIONS

Our work is limited in that it only involves the referential game. Another popular variant, the reconstruction game (e.g., Chaabouni et al., 2019a; 2020a; Lian et al., 2021; Conklin & Smith, 2023), requires the listener to reconstruct the input object based on the speakers' message. This setup may present different learning biases and thus have different results. We still expect inter-agent representational alignment to happen while losing connection to the image embeddings since there is no pressure to retain the latter connection. It would, however, be interesting to investigate whether the language protocol in this scenario is more structured than in the referential game.

Another limitation in our setup is that we only consider the scenario with two agents, which may be a requirement for alignment to be possible. Since experiments with human participants show that larger communities create more systematic languages (Raviv et al., 2019b), simulations on emergent multi-agent communication with populations of agents are also conducted, but these mostly yield negative (i.e., the emergent language protocol is not more structured) results (Michel et al., 2023). Nevertheless, we believe that emergent communication with populations of agents is ecologically more valid and could result in different alignment effects since this introduces the pressure to communicate with more than one other agent.

The last potential limitation of our study regards its scale. While simulations of emergent communication are typically conducted on relatively small-scale datasets, human language acquisition is accompanied by rich and diverse multi-modal experiences. Recent results in the field of computer vision suggest that dataset diversity and scale are the primary drivers of alignment to human representations (Conwell et al., 2023; Muttenthaler et al., 2023). As such, this key difference between the setting of artificial emergent communication and human language acquisition can drive the observed differences in representations. Due to the difficulty of interpreting these representations, we see this as another reason to evaluate emergent protocols on more strict datasets with clear pragmatic value for humans.

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

## A  CHANNEL CAPACITY

To test to what degree communicative success, TOPSIM, and representational alignment are confounded with the communication channel capacity, we ran simulations altering the vocabulary size ($V = \{3, 5, 10, 20, 40, 50, 100\}$) and message length ($L = \{2, 3, 5, 10, 50, 100\}$) resulting in 42 parameter settings per loss type.

Overall, performance is relatively independent of the chosen configuration, but vocabulary size influences success more than message length (Figure 5). The hyperparameters that resulted in the best validation accuracy (i.e., generalisation; Chaabouni et al., 2022) for the standard $ce$ setup were $V = 40$ and $L = 2$. These are used in the main paper. Contra expectations, this is also true for TOPSIM, which, especially in the case of $ce + L_{\mathrm{RSA}}$, is higher when messages are shorter but have access to a larger vocabulary (Figure 6).

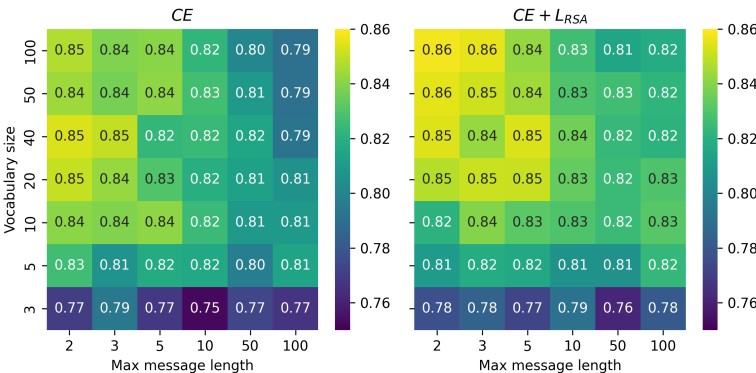

Figure 5: The validation accuracy as a dependent factor of the vocabulary size and maximum message length. Values are averages across 15 seeds.

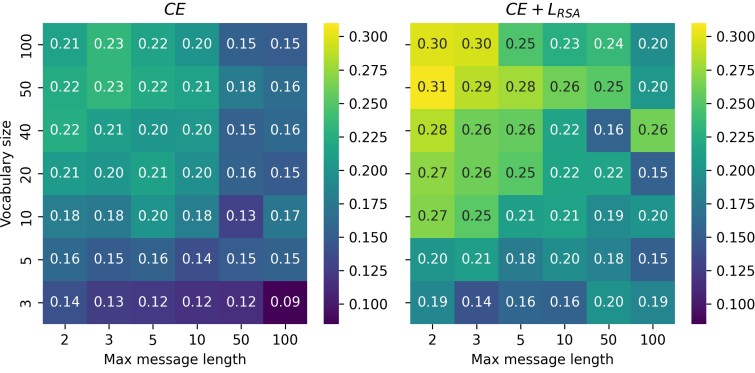

Figure 6: The validation TOPSIM as a dependent factor of the vocabulary size and maximum message length. Values are averages across 15 seeds.

Figure 7 shows that, regardless of capacity, inter-agent alignment increases while image-agent alignment decreases with the $ce$ loss. Interestingly, $\mathrm{RSA}_{sl}$ is agnostic to capacity but a larger vocabulary size, not message length, reduces the degree of drifting away from the input. We hypothesise this to result from lower pressure to compress rich continuous embeddings into smaller discrete vocabulary embeddings.

## B  BEST HYPERPARAMETERS

The parameters used to run our experiments were the following:

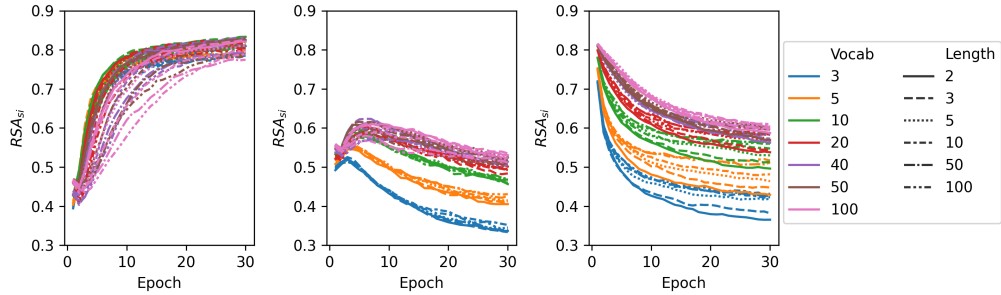

Figure 7: Representational alignment metrics averaged over 15 simulations with the standard *ce* loss. Representational alignment always occurs while losing relation to the input.

| Parameter | Value |
|---|---|
| Batch size | 32 |
| Optimiser | Adam |
| Learning Rate (S & L) | 0.01 & 0.001 |
| Vocabulary size ($V$) | 40 |
| Message length ($L$) | 2 |
| Hidden size (S & L) | 768 & 768 |
| Embedding size | 50 |
| Listener cosine temperature | 0.1 |
| Seeds | 16,22,41,56,67, 77,14,78,99,23, 82,40,51,37,62 |

Table 1: Best-performing parameters resulting from the parameter sweep that were used to obtain the main results.

## C  INTERACTION OF THE ALIGNMENT TERM ON THE CROSS-ENTROPY LOSS

To ensure that there is no impact of the alignment penalty on the pressure for communicative success, we ablated the $L_{\text{RSA}}$ term of our proposed loss function and found that both, communicative success and *ce* are not affected by the alignment penalty (Figure 8). Corroborating that only the *ce* term provides pressure for successful communication (§5.4).

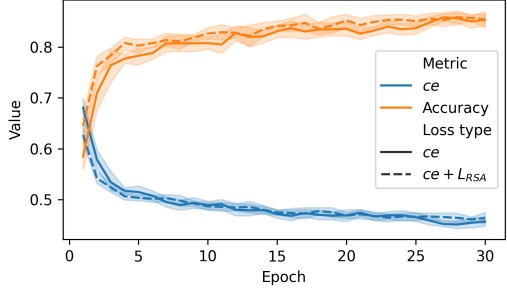

Figure 8: Learning curves (accuracy) and cross-entropy loss (*ce*) for both loss settings. There is virtually no effect of the auxiliary term $L_{\text{RSA}}$ on the cross entropy loss or communicative success.

## D  PRE-TRAINED VISION MODULES

Although it is in principle possible to train the vision module of the agents from scratch (Dessi et al., 2021), in our work, agents' perception stems from a pre-trained vision-language model.

Although we believe that DinoV2 embeddings capture high-level, conceptual image features useful for discriminating image pairs, we assessed the degree to which the alignment problem occurs for different pre-trained models despite encoding the same objects. We ran additional simulations using image features obtained from ResNet (He et al., 2016) and CLIP (Radford et al., 2021) for 6 different parameter settings with the $ce$ loss function. Here we used the parameters that resulted in the best, worst, mean, and quantile validation performance from the parameter sweep in appendix A (see Table 2), and a sensible setup with $V = 10$ and $L = 5$.

| Message Length ($L$) | Vocab. Size ($V$) | Vision |
|---|---|---|
| 2 | 40 | |
| 3 | 10 | |
| 5 | 5 | DinoV2 |
| 5 | 10 | CLIP |
| 10 | 3 | ResNet |
| 50 | 100 | |

Table 2: The parameters for running additional simulations with CLIP and ResNet to assess the robustness of our results. Each combination was run for 15 different seeds. Note: the results for the DinoV2 simulations are from the sweep.

Figure 9 shows clearly that inter-agent alignment *increases* while agent-image alignment *decreases* for all models. In addition to the similar results reported by Bouchacourt & Baroni (2018) for VGG ConvNet embeddings, both 4096 and 1000 layers, we can confirm that the problem is agnostic to the input embeddings. Interestingly, agent representations drift most for CLIP embeddings. Nevertheless, the agents still develop a successful communication strategy, indicating that out-of-the-box CLIP embeddings are the least useful for agents and enforce finding a different (non-grounded) solution. No such differences are seen when the agents are trained with the additional alignment penalty term, inter-agent and image-agent alignment remain high for all models.

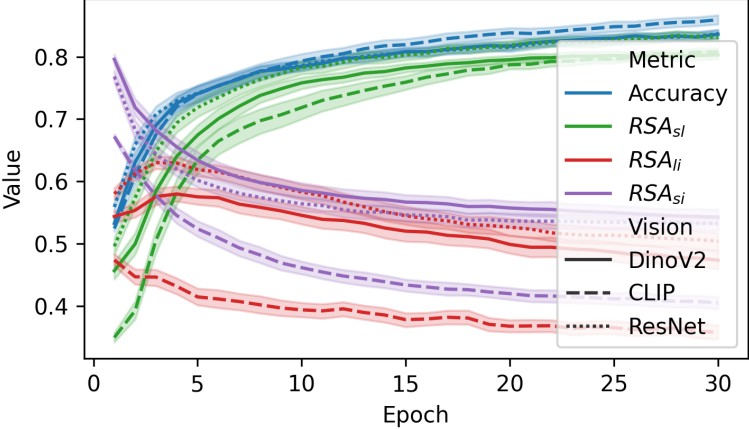

Figure 9: Learning curves (accuracy) and RSA metrics for different vision models averaged over 6 parameter settings with 15 seeds each. Line style corresponds to the vision module used to obtain image embeddings and colour indicates the metric. Areas indicate the 95% confidence intervals.

