# OpenReview forum: "The Curious Case of Representational Alignment: Unravelling Visio-Linguistic Tasks in Emergent Communication"
_ICLR.cc/2024/Workshop/Re-Align — ICLR 2024 Workshop Re-Align Poster_

### Official Review · Reviewer_PxD8 · 2024-02-21
**Very interesting results, some suggestions for the interpretation/framing**

**Rating:** 3
**Fit:** 3
**Confidence:** 3

**Workshop Review:**

This paper contains a thought-provoking set of experiments on emergent grounded communication settings. The authors train listener and speaker models to communicate and evaluate their generalization to out-of-distribution datasets (a challenging naturalistic one, and a pure noise one). The paper compares both accuracy and representational similarity across datasets. It notes low degrees of representational alignment despite success even on OOD datasets ins some cases. The paper proposes a similarity-structure alignment auxiliary loss to improve results, which does indeed yield greater representational alignment, but less dramatic changes to performance.  I think these results are quite compelling. However, I have a few comments on the interpretations thereof:
* First, it seems to me that achieving some accuracy on noise stimuli does not *necessarily* indicate the use of spurious features. For example, for the noise images shown in Fig. 2, a human might say something like "more cyan than red close to the bottom right corner" to indicate which image to select, and likely would have decently high success with a strategy  like this — even though we are using conceptual rather than spurious features. (Of course, the exact utterance given here relies on the comparison to the distractor, whereas I assume that as in standard EC setups the speaker only sees the target, not the distractors, but slightly longer utterance could likely work without needing a comparison as it's very unlikely that noise images will match in many different salient features.)
* I wonder to what extent the results are driven by the scale of the datasets used for emergent communication. For example, various recent results suggest that in computer vision, dataset diversity and scale is the primary driver of alignment to human representations (e.g. https://www.biorxiv.org/content/10.1101/2022.03.28.485868v2 or https://arxiv.org/abs/2211.01201). Given those results, it might be worth discussing this factor as one of the key differences between the setting of artificial emergent communication, and the rich experiences of human language.
    - (in addition to other factors of course, like a population of interlocutors rather than a single pair, which I appreciated the mention of in the limitations.)
* At the highest level, these results seem to be part of a larger story that the relationship between representation and computation is not as simple as we'd like it to be (see, for example, multiple realizability: https://plato.stanford.edu/entries/multiple-realizability/). Therefore, the representation structure we *expect* to help solve a task will not necessarily. (For example while disentangled representations are often positited to be beneficial for generalization, that isn't necessarily true in many cases; e.g. https://proceedings.neurips.cc/paper_files/paper/2022/hash/9f9ecbf4062842df17ec3f4ea3ad7f54-Abstract-Conference.html or https://openreview.net/forum?id=qbH974jKUVy) Similarly, I'm not sure that it's obvious *a priori* that having aligned representational structures between the speaker and listener would be either necessary or sufficient for improved communication performance or focus on more conceptual features — though there might be some reason to think it's likely to be useful. I think the work would benefit from explicitly bringing this general theme (of representation being indirectly linked to computation) out more explicitly, and the workshop/field might also benefit from seeing this work as an example of such.

**Reason For Not Giving Higher Score:**

N/A

**Reason For Not Giving Lower Score:**

Clear and interesting experiments, important demonstration of the subtleties of the relationship between representation and computation.

**Reviewer Domain:**

cognitive science

---

### Official Review · Reviewer_JtHn · 2024-02-24

**Rating:** 2
**Fit:** 3
**Confidence:** 1

**Workshop Review:**

The paper explores the emergence of linguistic properties in natural language and artificial agents. It discusses the complexity of language compositionality and the challenges in simulating this in deep neural agents in referential games. It suggests that a key issue with current simulations is the lack of alignment between the agents image representations and the actual images. This misalignment results in an artificial language that does not encode visual features. To address this, the authors propose introducing an alignment penalty. While this improves communication, it does not lead to a language grounded in images, highlighting the significant differences between human and artificial language emergence. Finally the paper underscores the importance of understanding these discrepancies to improve simulations of language emergence.

the paper is well written and clear, the findings are engaging and relevant to the community, and it fits well with the workshop theme, including also comprehensive sections on discussion and limitations.

**Reason For Not Giving Higher Score:**

The evaluation could have been enhanced by utilizing a wider variety of models or more recent ones.

**Reason For Not Giving Lower Score:**

The paper is well-structured and focuses on a particular context and assessment that is intriguing and valuable to the community.

**Reviewer Domain:**

machine learning

---

### Decision · Program_Chairs · 2024-03-02

Accept (Poster)